# Patient Reported Outcomes and Measures in Children with Rhabdomyosarcoma

**DOI:** 10.3390/cancers15020420

**Published:** 2023-01-09

**Authors:** Marloes van Gorp, Martha A. Grootenhuis, Anne-Sophie Darlington, Sara Wakeling, Meriel Jenney, Johannes H. M. Merks, Lisa Lyngsie Hjalgrim, Madeleine Adams

**Affiliations:** 1Princes Máxima Center for Pediatric Oncology, 3584 CS Utrecht, The Netherlands; 2School of Health Sciences, University of Southampton, Southampton SO17 1BJ, UK; 3Founder, Alice’s Arc, Rhabdomyosarcoma Children’s Cancer Charity, London E4 7RW, UK; 4Children’s Hospital for Wales, Cardiff CF14 4XW, UK; 5Department of Paediatrics and Adolescent Medicine, Rigshospitalet, 2100 Copenhagen, Denmark

**Keywords:** childhood cancer, rhabdomyosarcoma, patient-reported outcomes, patient-reported outcome measures, quality of life, adverse events

## Abstract

**Simple Summary:**

Children with rhabdomyosarcoma often experience difficulties as a result of their disease and treatment, such as pain or low mood. This can have a significant impact on their overall quality of life. It is important to evaluate these outcomes independently and prospectively, to improve the care provided for this population. One approach is to use questionnaires (or patient-reported outcome measures, PROMs) completed by patients. This commentary aims to encourage the use of PROMs by informing professionals in the field. The few available studies suggest that the quality of life of children with rhabdomyosarcoma is impaired. Additionally, children with rhabdomyosarcoma may have problems specifically related to their disease, for example due to their appearance after having surgery and/or radiotherapy. It is therefore important to develop questionnaires that include disease-specific issues. These can be used in addition to the generic quality of life questionnaires which are now more often used for children with cancer.

**Abstract:**

In addition to optimising survival of children with rhabdomyosarcoma (RMS), more attention is now focused on improving their quality of life (QOL) and reducing symptoms during treatment, palliative care or into long-term survivorship. QOL and ongoing symptoms related to the disease and its treatment are outcomes that should ideally be patient-reported (patient-reported outcomes, PROs) and can be assessed using patient-reported outcome measures (PROMS). This commentary aims to encourage PRO and PROM use in RMS by informing professionals in the field of available PROMs for utilisation in paediatric RMS and provide considerations for future use in research and clinical practice. Despite the importance of using PROMs in research and practice, PROMs have been reported scarcely in paediatric RMS literature so far. Available literature suggests lower QOL of children with RMS compared to general populations and occurrence of disease-specific symptoms, but a lack of an RMS-specific PROM. Ongoing developments in the field include the development of PROMs targeted at children with RMS specifically and expansion of PROM evaluation within clinical trials.

## 1. Background

In addition to optimising survival of children with cancer, attention has now turned to improving their quality of life (QOL) and reducing symptoms during treatment, palliative care or into long-term survivorship. QOL is a multidimensional subjective concept, including social, emotional, cognitive and physical functioning. QOL is affected by one’s health and is often used interchangeably with health-related quality of life (HRQOL) [1].

Rhabdomyosarcoma (RMS) is the most common soft tissue sarcoma in childhood, with varying local tumour extent and systemic stage, occurring in a range of anatomical disease sites. Treatment often comprises multiple modalities including chemotherapy, surgery and radiotherapy [2]. Because of this diversity, relevant symptoms or late effects that may impact QOL vary between subgroups of children with RMS. For example, children with head and neck RMS may experience facial asymmetry, hearing loss and pituitary dysfunction whereas those with pelvic tumours may have bladder and bowel dysfunction and a higher risk of infertility [3,4].

The monitoring of QOL in children with cancer has been recommended as a standard of psychosocial care [5], and there is a growing recognition of the need to include assessments of QOL in clinical trials to provide insight into the short and long-term impact of new treatments on how patients feel and function [6,7]. Additionally, in the care for children receiving palliative care, optimising QOL is the main aim [8]. The toxicity of the treatment for RMS can cause a range of symptoms, including for example, oral mucositis, vomiting and anxiety [9]. A patient’s self-reported perception is the best way to assess QOL and symptoms and can be captured using patient-reported outcomes (PROs) [10]. PROs are typically assessed using questionnaires, also called patient-reported outcome measures (PROMs). PROMS is therefore an umbrella term for instruments used to capture HRQOL domains (physical, social and psychological issues), functional status and subjective wellbeing [11]. Currently, PROM use in paediatric RMS care and research is relatively scarce and symptoms are often only graded by healthcare professionals, who base their assessment mostly on observation [12]. This is problematic, especially for less visible symptoms, because agreement between children’s, parent’s and clinician’s grading is low, and clinicians consistently underreport children’s symptoms [9,13]. All in all, it is time to systematically include the child’s voice in paediatric RMS care and research [14]. The aim of this commentary is to encourage PRO and PROM use in RMS by informing professionals in the field of available measures. This paper will first outline general considerations in utilising PROs and PROMs and provide an overview of PROs within the current literature in RMS specifically. The paper then outlines available PROMs, the use of PROs and PROMs in RMS trials and clinical practice, and finally future developments.

### 1.1. Considerations in Using PROs and PROMs in Paediatric RMS

When using PROMs, it is important to consider the method of assessment. In children with RMS the majority of assessments have been paper-pencil based so far. Although paper and pencil testing is low in costs, digital assessment reduces logistical steps with regard to data entry and processing for clinical use or research and reduces the risk of missing data [15], while outcomes are equivalent [16,17]. Web-based assessment increases the flexibility of digital assessment by allowing patients to report from home and improves the use of PROs in clinical practice [18].

There are multiple additional challenges to addressing PROs in paediatric RMS populations specifically. Firstly age: embryonal RMS is the most common major subtype of RMS and is most commonly diagnosed in children under five [2], who are unable to report outcomes themselves. Even though discrepancies are found between child and parent reported outcomes [10,19], agreement is higher than that between children and clinicians [9]. Therefore, for young children, the parent-report of PROMs may be the best solution to capture PROs until children are considered able to reliably complete a PROM. From the age of five years children should be able to self-report PROMs, if used in an interview format or with support from their parents. Children are usually able to independently self-report from eight years of age [20].

An additional challenge is that at different ages, different PROs or versions of PROMs are required as children develop their motor skills and emotional regulation, for example, and experience changes in their daily activities. The resulting differences between PROMs used at different time points can complicate (longitudinal) comparison. Similarly, subgroups with different RMS tumour locations may have different relevant outcomes depending on the site and extent of the original disease and its local treatment.

Finally, there are benefits and limitations to using generic or disease-specific PROMs. Generic PROMs allow comparison to other populations, most importantly the general population. However, generic PROMs sometimes do not capture disease-specific problems, such as those related to limb amputation or head and neck surgery or radiotherapy. Use of disease-specific PROMs may therefore be necessary, especially for clinical purposes where detailed information on individual functioning is required. In line with this, a recent review concluded that there is a need for a sarcoma-specific QOL instrument [21]. On the other hand, disease-specific PROMs may not have normative data available, because the measures do not apply in the absence of the effects of disease (e.g., questions relating to phantom pain or scars). In most cases, the combination of generic and disease-specific measures is likely to yield the most useful information.

### 1.2. PROs in Childhood Cancer and (Paediatric) Sarcoma

PROs are increasingly applied to capture the subjective experience of childhood cancer patients. The tools used are usually questionnaire based and encompass specific domains including HRQOL domains and functional status [22]. Childhood cancer diagnoses and treatment are greatly distressing and negatively impact the quality of life (QOL) of these children. Additionally, long-term survivors of childhood cancer are known to have lower overall QOL scores than their peers [23]. The use of PROs in paediatric sarcoma have generally been limited to measuring long-term outcomes. It is well documented that adult survivors of childhood cancer have lower HRQOL scores than their peers and higher rates of chronic health conditions and all-cause mortality [24,25]. Most studies are retrospective and use generic PROMS, and prospective data is currently limited to studies of HRQOL in adult sarcoma patients [26,27]. A recent systematic review in the adult sarcoma population has demonstrated that sarcoma patients scored lower in physical and psychological QOL domains and experienced higher rates of self-image issues, depression, and suicide when compared to healthy individuals [21]. Although scoring is similar to other cancer patients, the lack of sarcoma-specific measures led to concerns that key issues may have been omitted [21].

### 1.3. PROs in (Paediatric) RMS Literature

Although there have been calls in the literature on children with RMS to increase QOL reporting [15], reports of PROs remain very limited. Large cohort studies on long-term survivors of childhood cancer have found survivors of soft tissue sarcoma (of whom the majority had a RMS diagnosis) had lower overall physical health-related quality of life compared to other survivors of childhood cancer [28]. Compared to general population controls, long-term survivors of soft tissue sarcoma from the St Jude Lifetime Cohort Study had lower QOL on all domains and lower social attainment (e.g., they less often had full-time employment or lived independently) [29]. Similarly, even though a majority of adult survivors of paediatric RMS from the Childhood Cancer Survivor Study completed high school and got married, they were less likely than siblings to achieve such milestones [30]. Finally, health problems occurred often in these RMS survivors, of whom 85% had at least one medically diagnosed condition [30].

In a few smaller and more selective RMS subgroups PROs have been used mostly to capture diagnosis-specific (e.g., head and neck RMS) or treatment-specific (e.g., proton therapy) outcomes. Long-term survivors of head and neck RMS had relatively similar overall QOL when compared to the general population. However, disease-specific consequences were found: for example, over half of these survivors negatively reported on their appearance [31]. In survivors of bladder-prostate RMS and genital tract tumours, PROs were used to evaluate urinary, digestive and sexual function and overall QOL [32,33]. Finally, some studies used PROs to evaluate treatment. In RMS patients treated with pencil beam scanning proton therapy, QOL was reduced in most domains at the start of treatment, which had returned to levels similar to those of the general population in most domains two years after treatment [34]. Similarly, patients with spinal RMS had less pain after surgery compared to before [35].

### 1.4. Available PROMs and PROMs in Development

A recent review has demonstrated >15 generic PROMs (e.g., the Pediatric Quality of Life Inventory Core Module [PedsQL™]) and a similar number of cancer-specific tools (e.g., PedsQL cancer module) in childhood cancer [22]. These measures comprise varying domains designed to capture physical health (e.g., function and symptoms such as pain and fatigue), psychological health (e.g., anxiety and body image), and social health (e.g., relationships and school attendance).

The development of a PROM may seem relatively easy, but it requires a thorough and often lengthy process to ensure it is valid and reliable [36,37]. The PROM needs to be reflected through relevant items that should be developed based on the available literature and directly informed by stakeholders’ (e.g., clinicians, patients and parents) experiences [12,36,38,39]. The next step is to determine the validity of the PROM; i.e., how well does the PROM assess the patient’s experience of the desired outcome? Further psychometric research should also be conducted to assess the reliability, i.e., does the PROM provide consistent results? Additional psychometric analyses are needed to optimise its potential in clinical use (e.g., what are clinically relevant cut-offs or what is the minimal important change according to patients?) and longitudinal research (e.g., the responsiveness, or: can the PROM detect change over time?) [40,41,42,43,44,45]. After development of an original version, translation to other languages and additional psychometric research for these translated versions is required [46]. Finally, for generic PROMs, general population norm data should be collected and made available.

We describe three well-known and available generic PROMs, which sometimes have relevant disease-specific modules, and one sarcoma-specific PROM. All of them are free to use for academic clinical trials, subject to a user agreement. There may be a fee for use in commercial trials.

#### 1.4.1. PedsQL™

The PedsQL™ [47,48,49] is the commonest PRO used in paediatric clinical trials [6]. This measure employs a modular approach to measuring QOL in healthy children and adolescents as well as those with acute and chronic health conditions. It comprises a brief (23 item) generic core scale as well as disease (including childhood cancer)-specific modules. It is often selected due to its brevity, the availability of age-specific versions including child and adolescent self-reporting (from age 5–18 years) as well as parent-proxy report (from infant to adulthood) and the fact that it has been translated and validated in many different languages. Young adult versions have also been developed more recently, allowing consistency in long-term follow-up of QOL (up to age 25) [50].

#### 1.4.2. PROMIS©

The Patient-Reported Outcomes Measurement Information System (PROMIS) is the second most commonly used PRO in paediatric oncology clinical trials [6], but has been rarely used in the RMS population so far. The PROMIS database comprises over 300 generic measures of QOL. After the initial development of instruments for adults [51], PROMIS have produced paediatric instruments [52] that include self-reports for children aged 8–17 years and parent-proxy reports for 5–17 years. The English-language paediatric instruments were validated in children with cancer [53]; available translations and validation in other languages differ by outcome and are ongoing. Recently, parent-proxy early childhood instruments were developed to assess QOL in 1–5 year-olds [54]. The PROMIS database has over 600 questions covering many aspects of physical, mental and social health, which are combined to form specific measures. Measures include item banks, short forms and computerised adaptive testing (CATs), which can all be scored on the same metric. Item banks are the total collections of items for a PRO. Short forms are fixed-length forms comprising items selected from the item banks. To apply CATs, items are selected from an item bank based on a respondent’s answers to previous questions, which reduces the number of items needed (and thus reduces the burden of patients and/or parents) and improves the reliability of the result. PROMIS profiles combine short forms or CATs of common domains. An additional advantage of the PROMIS database over other generic PRO measures is that it allows flexibility to focus on certain aspects of QOL [55].

#### 1.4.3. EORTC-QLQ-C30

The European Organisation for Research and Treatment of Cancer (EORTC) is a large collaborative network of cancer research and clinical trials. The EORTC has a large Quality of Life working group which actively develops and refines questionnaires for assessing QOL in oncology clinical trials and also collaborates with disease-specific working groups to ensure QOL assessments are included in trial programs. The EORTC-QLQ-C30 is the core questionnaire validated for use in adult patients aged >18 years [56], and has been translated into 110 languages. An adolescent and young adult measure is in development for patients aged 14–39 years.

EORTC has disease-specific modules to complement the core questionnaires. For example, a sarcoma-specific module is being designed to be used alongside the EORTC-QLQ-C30 core questionnaire for adult patients aged 18 years and over. Additionally, a symptom measure for synovial sarcoma was recently developed [57,58]. The initial item generation is complete with the project currently in the item reduction stage [59]. To reduce items, an ongoing study plans to include 630 patients and health-care professionals internationally who will rate and prioritise the items on relevance. Although not currently in progress, a future development may be that a paediatric sarcoma-specific module of the EORTC will be developed as well, using the experience of the development of the adult module.

EORTC also has an item library comprising over 900 items which can be used alongside the core and disease-specific modules to allow inclusion of relevant QOL issues not captured by the modules [60]. The advantage of pre-validated item banks is that they allow the user to capture a wider range of symptoms and events without having to go through the time-consuming process of questionnaire development and validation.

#### 1.4.4. Sarcoma Assessment Measure (SAM)

The Sarcoma Assessment Measure is a 22-item questionnaire that has been recently developed to capture common experiences of all patients with sarcoma using standard mixed methodology (item generation, reduction and validation) [61]. The development study recruited 121 sarcoma patients from the United Kingdom, and the SAM questionnaire has been validated for patients aged >17 years. Currently, a UK-based study is using the same methodology to develop a paediatric version of the SAM questionnaire (SAM-Paeds) [62]. This will allow longitudinal monitoring of QOL with patients able to complete the paediatric version, followed by the adult version as they transition into long-term follow-up. This study is in progress, and at the time of writing the researchers have just completed the item reduction stage.

## 2. PROs in Clinical Trials

PROs can be very important in assessing the risk: benefit ratio of clinical trial interventions, in addition to traditional outcome measures such as overall and event-free survival or tumour response (e.g., objective response rate) [63]. Key aspects of QOL are complex and highly subjective (e.g., physical, social and emotional functioning) but critical in understanding the impact of new agents and therapies on patients’ well-being. Therefore both the Food and Drug Administration and the European Medicines Agency have recommended the use of PRO measures to support outcome reporting in paediatric oncology clinical trials [39,64]. Despite these recommendations, an underutilisation of PROs in trials is well documented, and PROs are used as end points in only 8.2% (6% as primary outcomes) of cancer trials involving children and adolescents < 21 years old [6]. Even in trials investigating supportive interventions to improve QOL, less than half used PROMs [65]. There are several reasons for this poor uptake of PROs, including the cost of trial development and the need to change the culture of clinical trials to prioritise patient outcomes. Consequently, there are now recommendations for the systematic education of trial team members on PRO research and to involve multidisciplinary and patient stakeholders in the comprehensive process of PRO assessment within clinical trials [63,66]. Moreover, to improve the use of PROs in trials, guidelines and international standards have been developed [67,68,69].

### FAR-RMS Clinical Trial

The Frontline and Relapsed RMS (FAR-RMS) clinical trial (ClinicalTrials.gov Identifier: NCT04625907) is an overarching study for newly diagnosed and relapsed RMS, led by the European paediatric Soft tissue sarcoma Study Group (EpSSG) [70]. This multinational trial began recruitment in 2020 and has a number of chemotherapy and radiotherapy randomisations.

QOL is a primary outcome measure in this trial, with specific focus on the radiotherapy randomisations: pre vs. post-operative radiotherapy for patients with localised, resectable disease; dose escalation for patients with high-risk unresectable tumours and limited vs. extensive radiotherapy for patients with widespread metastatic disease. QOL scores will be used, alongside survival outcomes and toxicity data to make recommendations for gold-standard strategies going forward. If survival outcomes are equivocal between radiotherapy randomisation groups, the toxicity and QOL data will be vital to interpret which strategies to take forward as standard of care in subsequent clinical trials.

QOL assessment in this trial comprises PedsQL™ for paediatric patients and EORTC-QLQ-C30 for adult patients, and data are collected up until two years after completion of local therapy. The SAM-Paeds measure will be included as an amendment once the questionnaire is finalised with the aim of increasing the sensitivity and specificity of results. An expansion to this study protocol to further evaluate QOL in all study patients is now in draft with the intention of increasing the duration of QOL follow-up to 10 years or age 20, whichever is the longer, to gain insight in late effects.

## 3. PROMs in Clinical Practice

### 3.1. Process of Implementation

To use PROs for individual patients in clinical practice, several steps need to be taken [71]. The multidisciplinary team of healthcare professionals needs to decide on the most appropriate, valid and reliable PROMs for their patient and on how they can implement them across the patient’s clinical pathway. Patients need to be involved in this process, ideally co-designing the plan for the use of PROMs, including consideration for patients with a specific condition and age ranges [72]. Thereafter, patients can complete the selected PROMs before a consultation with the clinician, preferably online from the comfort of their home. Digital completion allows visualisation of the PROM results in a dashboard, which can be made available to the clinician and patient before and during the consultation [15]. The clinician is ideally trained in how to interpret and use the PROM data in clinical practice [73], and discusses the PROM outcomes with patients, and parents if present, during consultation. In this way, they can monitor functioning over time, screen for and identify any problems and subsequently provide tailored advice and interventions, or refer to the appropriate psychosocial support within an appropriate time frame [74].

### 3.2. Effectiveness of PROMS in Clinical Practice

Over the years many studies have investigated the effect of the use of PROMs in clinical practice. These studies showed that using PROMs in adult oncology clinical practice increased the discussion with patients relating to outcomes, enhanced patient-clinician communication, resulted in higher patient satisfaction, improved patient outcomes such as QOL, and even improved survival [75,76,77]. In a study on children with cancer there was improved discussion with better detection of problems, higher satisfaction with care, increased treatment engagement and enhanced patient-clinician communication [78].

### 3.3. Barriers to Implementation

Barriers to implementation can be found on several levels: the intervention (e.g., suboptimal fit with current routines, inability to access PROMS or relevant online platforms), the users (e.g., lack of support from the multidisciplinary team), and the organisation (e.g., financial barriers) [79]. In an international survey, healthcare professionals reported that some barriers for assessing PROMS were: time (58%); insufficient staff (49%), logistics (32%) and financial resources (26%) [80]. Providers from developing countries more often reported barriers concerning insufficient staff, logistics and financial resources. Patients and parents reported that PROMs were sometimes irrelevant and repetitive, and that they were not satisfied by the extent to which the PROMs were discussed by their clinician [81].

## 4. Future Directions

### 4.1. Technology

Questionnaire-based PROs are limited by being subjective, often retrospective and relying on patient recall. Digital health interventions may be a useful tool to measure real-time outcomes such as mobility, self-reported symptoms and QOL scores. Digital health interventions, defined as ‘the use of mobile and wireless technologies for health to improve health system efficiency and health outcomes’, provide the opportunity to connect patients to health care providers and clinical trial coordinators in real time [82]. Examples of digital health interventions include wearable activity trackers, mobile phone applications and real-time assessment of PROMs. Digital health interventions are most widely reported in adult surgical patients, for example to track post-operative recovery [83]. Digital health interventions have been used in paediatric studies, such as improving education in patients with asthma [84], or to monitor pain in children with cancer [85], but they are not yet widely implemented in paediatric RMS patient care. A proposal for a pilot study is in development.

### 4.2. International PROM Platform

To facilitate the uptake and potential impact of PROs in clinical trials it is necessary to collect, store, and analyse PROMs in an efficient and consistent manner, so data can be compared across institutions and countries both to increase sample sizes and enhance understanding of the impact of the clinical interventions on patients. An international online questionnaire portal could provide such opportunities. Benefits of using an online portal include reducing the burden of completion for patients and families, as the same PROMs can be used across clinical care and within clinical research. Also, advanced methods such as computerised adaptive testing are only possible through this type of centralised, online format. Whilst the need for such an international platform is well-recognised in the literature as well [2], its realisation is complicated by different data protection regulations and costs.

## 5. A Parent Advocate’s Perspective

QOL is a key factor that should be at the forefront of interactions between clinicians, their patients and the patients’ parents. It is critical to consider QOL throughout the treatment pathway to inform decision-making around treatment options and life beyond treatment. To do this effectively, quality data needs to be captured, analysed, and reported in a user-friendly manner and made accessible to the patient/parent community. Disease-specific PROM’s have the potential to achieve this. Although parents and patients are often interviewed and asked to trial the questionnaires as part of PRO development, it is not always clear whether there has been patient and public involvement (PPI) at all stages of new PRO design. This is an important factor to consider in future studies to ensure PPI is used in the creation of PROMS as standard policy.

Parents and patients can play a vital role in utilising PROM’s to ensure more attention is given to QOL in care for children with RMS and in clinical trials such as FaR-RMS. This could be done by helping to devise the PROM questions; ensuring the PROM uses appropriate language; encouraging participation in PROMs; reviewing results with clinicians; preparing user-friendly communications; summarising the results; disseminating findings effectively; demonstrating the value of PROMs, and soliciting qualitative feedback from participants.

## 6. Conclusions

This commentary paper aimed to inform professionals on the use of PRO and PROMS in the setting of children with RMS. Despite the clear value of using PROs and PROMs in research, they have been reported scarcely in paediatric RMS literature to date. The limited available literature suggests that the experience of having RMS and its treatment in childhood results in lower QOL scores when compared to the general population as well as the presence of disease-specific symptoms. Ongoing developments in the field include development of PROMs designed to be used for children with RMS and an expansion of PROM evaluation within current clinical trials.

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
