# Peer review of "Patient Reported Outcomes and Measures in Children with Rhabdomyosarcoma"

_cancers, 2023, doi:10.3390/cancers15020420_

Round 1
Reviewer 1 Report
Reviewer comments
Patient reported outcomes and measures in children with rhabdomyosarcoma 3
Marloes van Gorp et al
Overall comments:
1. This is a brief review on a very important topic. Unfortunately, the authors have failed to give the issue the necessary depth. Too often their review gets stuck in wordy claims of the obvious: we do need to listen to our patients. The matter for the review should be more focused: how do we best listen? When it comes to the title: how do we best listen to children with rhabdomyosarcoma?
2. The authors generalise, where practical and important information is available to shape the context. Whilst they demonstrate how sarcoma location and therapy choice matters, they fail to make reference to the many extant adaptable tools for regionalised and generalised sequelae of sarcoma and sarcoma therapy. Some of these issues are listed in the proposed references below, but all over we propose to go back to the drawing board and give this review the depth and detail it deserves.
3. The manuscript is reasonably structured, but could be much improved by a more focused and pertinent phrasing. This matters – as this is a review paper on a rather complex matter where all hinges on the clarity of communication.
Just an example: line 52: there is a growing need to include; authors mean: there is a growing recognition of the need to include; or 56: the patient’s voice is considered the gold standard; authors mean: self-reported perception of their health and well-being are the best way to assess QoL
Not rarely, phrases are syntactically incorrect/incomplete: line 82: “PROMs may be the best alternative until children are considered able to reliably self-report a PROM, which appears to be from 5 years”
----------------------------------------------------
Recommended literature for inclusion:
Patient reported quality of life in young adults with sarcoma receiving care at a sarcoma center.
Day JR, Miller B, Loeffler BT, Mott SL, Tanas M, Curry M, Davick J, Milhem M, Monga V.
Front Psychol. 2022 Sep 29;13:871254. doi: 10.3389/fpsyg.2022.871254. eCollection 2022.
PMID: 36248560
Free-living monitoring of ambulatory activity after treatments for lower extremity musculoskeletal cancers using an accelerometer-based wearable - a new paradigm to outcome assessment in musculoskeletal oncology?
Furtado S, Godfrey A, Del Din S, Rochester L, Gerrand C.
Disabil Rehabil. 2022 Jun 16:1-10. doi: 10.1080/09638288.2022.2083701. Online ahead of print.
PMID: 35710327
Qualitative study to characterize patient experience and relevance of patient-reported outcome measures for patients with metastatic synovial sarcoma.
Eliason L, Grant L, Francis A, Cardellino A, Culver K, Chawla SP, Arbuckle R, Pokras S.
J Patient Rep Outcomes. 2022 May 4;6(1):43. doi: 10.1186/s41687-022-00450-1.
PMID: 35507231
Equivalence of paper and electronic-based patient reported outcome measures for children: a systematic review.
Kortbeek S, Pawaria A, Ng VL.
J Pediatr Gastroenterol Nutr. 2022 Oct 14. doi: 10.1097/MPG.0000000000003636. Online ahead of print.
PMID: 36240491
Psychometric properties of patient-reported outcomes measures used to assess upper limb pathology: a systematic review.
Abbot S, Proudman S, Sim YP, Williams N.
ANZ J Surg. 2022 Aug 12. doi: 10.1111/ans.17973. Online ahead of print.
PMID: 35959939 Review.
Patient-reported outcomes are under-utilised in evaluating supportive therapies in paediatric oncology - A systematic review of clinical trial registries.
Rothmund M, Lehmann J, Moser W, de Rojas T, Sodergren SC, Darlington AS, Riedl D.
Crit Rev Oncol Hematol. 2022 Aug;176:103755. doi: 10.1016/j.critrevonc.2022.103755. Epub 2022 Jul 5.
PMID: 35803454 Free article. Review.
Quality of patient- and proxy-reported outcomes for children with impairment of the upper extremity: a systematic review using the COSMIN methodology.
Kalle JPR, Saris TFF, Sierevelt IN, Eygendaal D, van Bergen CJA.
J Patient Rep Outcomes. 2022 Jun 2;6(1):58. doi: 10.1186/s41687-022-00469-4.
PMID: 35652989 Free PMC article. Review.
Patient-reported outcome measures and value-based medicine in paediatrics: a timely review.
Tan YH, Siew JX, Thomas B, Ng KC.
Singapore Med J. 2021 Sep 21. doi: 10.11622/smedj.2021102. Online ahead of print.
PMID: 34544213
Correlation Between the PROMIS Pediatric Mobility Instrument and the Hospital for Special Surgery Pediatric Functional Activity Brief Scale (HSS Pedi-FABS).
Adjei J, Schachne JM, Green DW, Fabricant PD.
HSS J. 2020 Dec;16(Suppl 2):311-315. doi: 10.1007/s11420-019-09726-7. Epub 2019 Dec 4.
PMID: 33380962
Patient-Reported Outcome Measures in Routine Pediatric Clinical Care: A Systematic Review.
Bele S, Chugh A, Mohamed B, Teela L, Haverman L, Santana MJ.
Front Pediatr. 2020 Jul 28;8:364. doi: 10.3389/fped.2020.00364. eCollection 2020.
PMID: 32850521
Patient-reported outcome measures in pediatric palliative care-a protocol for a scoping review.
Holmen H, Winger A, Steindal SA, Castor C, Kvarme LG, Riiser K, Mariussen KL, Lee A.
Syst Rev. 2021 Aug 28;10(1):237. doi: 10.1186/s13643-021-01791-6.
PMID: 34454605
Line 46: the main diversity is regional tumour extent and systemic stage, to a degree predicting treatment approach; please rephrase
Author Response
Please see attachment with each point addressed and text references of changes made

Reviewer 2 Report
The Authors aimed to review on the use of patient reported outcomes for quality of life in patients affected by rhabdomyosarcoma.
The title is misleading. Also, both the summary and abstract can be misleading. It is not clear the narrative nature of this review.
The aim is not clearly reported also. Please detail.
Moreover, why did the Authors include only rhabdomyosarcoma? Is this topic different in non-rhabdomyosarcoma histotypes?
Maybe, this issue may differ according to the site of the tumor.
Something should also be discussed about age of children.
Author Response
Please find attached document with response to each point and text references of changes made

Round 2
Reviewer 1 Report
Dear Authors,
thank you for making a number of relevant changes. This paper much better sits as a commentary then a full paper and will now convey its message!
Happy Christmas days
Author Response
Thank you very much for taking the time to review the paper. We are very happy that you are pleased with the revisions.
Reviewer 2 Report
Despite the attempts to Ameliorate the paper, it still lacks of novelty. Also, it is very hard to understand what this paper can add to the existing Literature.
It is still not clear in what RMS differs from NRMS.
Author Response
Thank you very much for taking the time to review the paper.
We have tried to provide overview of current topics which are relevant to the subject of using and developing PROMS in paediatric oncology, with particular focus on rhabdomyosarcoma (over other sarcoma subtypes) due to the context of the paper within the rhabdomyosarcoma-specific special issue of the journal.
We appreciate that summarising the current use of PROMS and what is needed to make future steps might not be considered a novelty, but we feel it highlights the importance of this area in particular for paediatric oncology health care providers who are not experienced in this field. We hope that the paper can be considered essential in the context of the rhabdomyosarcoma special issue.
The document has been checked and revised to ensure that all of the punctuation and double-wording issues have been rectified.